# Novofumigatonin biosynthesis involves a non-heme iron-dependent endoperoxide isomerase for orthoester formation

Yudai Matsuda [1,2,3], Tongxuan Bai[2], Christopher B.W. Phippen[1], Christina S. Nødvig[1], Inge Kjærbølling[1], Tammi C. Vesth[1], Mikael R. Andersen [1], Uffe H. Mortensen [1], Charlotte H. Gotfredsen [4], Ikuro Abe [2] & Thomas O. Larsen [1]

Novofumigatonin (**1**), isolated from the fungus *Aspergillus novofumigatus*, is a heavily oxygenated meroterpenoid containing a unique orthoester moiety. Despite the wide distribution of orthoesters in nature and their biological importance, little is known about the biogenesis of orthoesters. Here we show the elucidation of the biosynthetic pathway of **1** and the identification of key enzymes for the orthoester formation by a series of CRISPR-Cas9-based gene-deletion experiments and in vivo and in vitro reconstitutions of the biosynthesis. The novofumigatonin pathway involves endoperoxy compounds as key precursors for the orthoester synthesis, in which the Fe(II)/α-ketoglutarate-dependent enzyme NvfI performs the endoperoxidation. NvfE, the enzyme catalyzing the orthoester synthesis, is an Fe(II)-dependent, but cosubstrate-free, endoperoxide isomerase, despite the fact that NvfE shares sequence homology with the known Fe(II)/α-ketoglutarate-dependent dioxygenases. NvfE thus belongs to a class of enzymes that gained an isomerase activity by losing the α-ketoglutarate-binding ability.

[1] Department of Biotechnology and Biomedicine, Technical University of Denmark, Søltofts Plads, 2800 Kgs. Lyngby, Denmark. [2] Graduate School of Pharmaceutical Sciences, The University of Tokyo, 7-3-1 Hongo, Bunkyo-ku, Tokyo 113-0033, Japan. [3] Department of Chemistry, City University of Hong Kong, 83 Tat Chee Avenue, Kowloon, Hong Kong SAR, China. [4] Department of Chemistry, Technical University of Denmark, Kemitorvet, 2800 Kgs. Lyngby, Denmark. Correspondence and requests for materials should be addressed to Y.M. (email: ymatsuda@cityu.edu.hk) or to I.A. (email: abei@mol.f.u-tokyo.ac.jp) or to T.O.L. (email: tol@bio.dtu.dk)

Orthoesters are functional groups in which three alkoxy groups are bound to a single carbon atom. Orthoesters are widely distributed in plant metabolites[1], but are also found in biologically important bacterial natural products, such as the sodium channel blocker tetrodotoxin[2] and the orthosomycin antibiotics[3]. Despite their intriguing structures and biological activities, little is known about the molecular basis for orthoester biogenesis. To the best of our knowledge, the Fe(II)/α-ketoglutarate (αKG)-dependent dioxygenases involved in the biosynthesis of orthosomycins are the only examples of orthoester-producing enzymes known to date[4].

Novofumigatonin (1) is a fungal meroterpenoid discovered in *Aspergillus novofumigatus* IBT 16806 (CBS117520)[5], and it possesses a congested skeleton with an unusual orthoester moiety. It has been proposed that 1 is biosynthesized via asnovolin A (2), which was also isolated from the same *A. novofumigatus* strain (Fig. 1a)[6]. Asnovolin A (2) is in turn derived from the polyketide precursor 3,5-dimethylorsellinic acid (DMOA, 3), which also serves as the starting material for diverse fungal meroterpenoids[7]. We recently identified a putative biosynthetic gene cluster for 1 (the *nvf* cluster; Fig. 1b, Supplementary Figs. 1, 2 and Supplementary Table 1)[8], but no biosynthetic study of 1 has been reported. Therefore, the biosynthetic pathway of 1, including the mechanism for the orthoester formation, has yet to be elucidated.

Here we show the complete biosynthetic pathway of novofumigatonin (1) and characterization of key enzymes for the orthoester synthesis. We initially create a series of gene-deletion strains of *A. novofumigatus* and isolated several metabolites, which we incorporated into a model for the biosynthetic route of 1 (Fig. 2a). The proposed pathway is further validated by heterologous expression experiments using *Aspergillus oryzae* and by in vitro enzymatic reactions with purified enzymes. In the course of our study, we achieve the total biosynthesis of 1 in the heterologous host *A. oryzae*, strongly indicating that the *nvf* cluster genes are sufficient to produce 1. We reveal the involvement of two key enzymes for the orthoester synthesis, namely NvfI and NvfE, which are responsible for the endoperoxidation and endoperoxide isomerization, respectively, to afford the orthoester. Most importantly, NvfE is characterized to be a non-heme iron-dependent endoperoxide isomerase despite its sequence similarity to known Fe(II)/αKG-dependent dioxygenases.

## Results

**Creation of an *A. novofumigatus* mutant for gene deletion.** To characterize the functions of each gene in the *nvf* cluster, we first created a mutant strain of *A. novofumigatus* in which genetic modifications can be readily performed (Supplementary Figs. 3, 4). To this end, the *pyrG* and *ligD* genes encoding an orotidine 5′-phosphate decarboxylase[9] and a DNA ligase were deactivated and deleted, respectively, by means of the recently developed fungal CRISPR-Cas9 system[10]. The *pyrG* gene serves as a selectable/counter-selectable genetic marker, and the disruption of *ligD* reportedly increases the gene-targeting efficiency due to non-functional non-homologous end-joining[11], which, in most cases, enabled efficient gene deletion without further usage of the CRISPR-Cas9 system and thus minimized vector construction efforts in the following experiments. With this mutant strain (*pyrG*⁻, *ligD*Δ) in hand, we initially deleted the polyketide synthase (PKS) gene *nvfA* (Protein ID in JGI database: 455569; see Supplementary Table 1 for details), which is expected to be responsible for the first committed step in the novofumigatonin biosynthesis. The deletion of *nvfA* completely abolished the production of 1 as well as 2 (Fig. 2b, lanes i and ii), indicating the involvement of the *nvf* cluster in the novofumigatonin pathway and strongly supporting the view that 2 is a precursor of 1.

**Early-stage biosynthesis of 1.** Since the initial biosynthetic steps of DMOA-derived meroterpenoids adopted similar pathways to afford their first cyclized intermediates[7], four gene products, the prenyltransferase NvfB, the methyltransferase NvfJ, the flavin-dependent monooxygenase (FMO) NvfK, and the terpene cyclase NvfL, as well as the PKS NvfA, were expected to be required for the formation of a tetracyclic compound, by analogy to the biosyntheses of austinol, terretonin, and andrastin A (Supplementary Fig. 5)[12–15]. First, the involvement of NvfB in the biosyntheses and its function as a DMOA farnesyltransferase were confirmed by the deletion of *nvfB* (Fig. 2b, lane iii) and heterologous expression of *nvfB* with the known DMOA-synthase gene *andM*[16] in the *Aspergillus oryzae* NSAR1 strain[17], which yielded farnesyl-DMOA (4) (Fig. 2d, lane ii and Supplementary Fig. 6).

Rather than the expected early intermediate, the *nvfJ*Δ strain accumulated the tetracyclic molecule 5, which was named asnovolin I (Fig. 2b, lane xi). This result surprisingly revealed that NvfJ acts after the cyclization event and therefore differs from the methyltransferases in the known DMOA-derived pathways, which require the methyl esterification of 4 for the activity of the terpene cyclases[7]. Consistent with this observation, the *nvfK*Δ and *nvfL*Δ mutants both yielded compounds 4 and 6 with a free carboxyl group, respectively (Fig. 2b, lanes xii and xiii). Compound 6 has a diol moiety at the terminus of the farnesyl group, which is likely to be a consequence of the hydrolysis of the expected epoxide 7. Collectively, the early-stage biosynthesis is proposed as follows. The PKS NvfA forms DMOA (3), followed by the farnesylation by NvfB, the epoxidation by NvfK, and the protonation-initiated cyclization of 7 by NvfL, to afford 8 via the cationic intermediate 9 (Fig. 2a).

**Mid-stage biosynthesis of 1.** In addition to the above-mentioned mutants, three more strains, *nvfC*Δ, *nvfH*Δ, and *nvfM*Δ, were unable to produce both 1 and 2 (Fig. 2b, lanes iv, ix, and xiv), and therefore these genes are likely to be engaged in the pathway to 2. Among the three mutants, the *nvfC*Δ strain did not accumulate any detectable unknown metabolites, and therefore the function of *nvfC* was investigated using the *A. oryzae* heterologous expression system; the transformant harboring *andM*, *nvfB*, *nvfK*, *nvfL*, and *nvfC* yielded both 8 and chermesin D (10)[18], while the transformant lacking *nvfC* only produced the alcohol form 8 (Fig. 2d, lanes iii and iv and Supplementary Fig. 6), establishing the function of NvfC as a 3-OH dehydrogenase of 8. Compound 8 was hereby designated as asnovolin H.

The *nvfH*Δ strain accumulated the product 11, which was named asnovolin J. The structure of 11 is nearly identical to that of 2, and only lacks the oxygen atom between C-3 and C-4, which established the FMO NvfH as a Baeyer–Villiger monooxygenase (BVMO). Another close analog of 2, 12, with a double bond between C-5′ and C-6′, was obtained from the *nvfM*Δ strain and designated as asnovolin K. This indicates that NvfM is an enoylreductase that reduces this double bond. Taken together, our results suggest that there are two pathways to synthesize 2 from 10. In one branch, 10 undergoes Baeyer–Villiger oxidation, methylation, and enoyl reduction to form 5, 12, and 2 in this order, while in the other branch, the methylation precedes the Baeyer–Villiger oxidation and the enoyl reduction to give 13, 11, and 2, respectively (Fig. 2a). Consistently, the introduction of the above-mentioned eight genes into *A. oryzae* successfully yielded 2 (Fig. 2d, lane v and Supplementary Fig. 6). In this proposed pathway, the methyl esterification by NvfJ occurs just before the NvfM-catalyzed reduction. Without the methylation, a β-keto acid that spontaneously undergoes decarboxylation would be generated upon the enoyl reduction. Thus, the methyl group could serve as a protecting group of carboxylic acid, which

explains the different timing of the methylation in the biosyntheses of **1** and the other known DMOA-derived meroterpenoids (Fig. 2a and Supplementary Fig. 5).

**Late-stage biosynthesis of 1**. Asnovolin A (**2**), but not **1**, was detected in extracts from five mutants, *nvfD*Δ, *nvfE*Δ, *nvfF*Δ, *nvfG*Δ, and *nvfI*Δ (Fig. 2b, lanes v to viii and x), suggesting that all five genes are involved in the synthesis of **1** from **2**. Except for the *nvfI*Δ mutant, all produced **2** in combination with another compound. The mutants without *nvfD* and *nvfE* accumulated the metabolites **14** and **15**, respectively (Fig. 2b, lanes v and vi). Compound **14** was also produced by the *nvfD*Δ, *nvfE*Δ, *nvfF*Δ, *nvfG*Δ quadruple-mutant strain (Fig. 2b, lane xv), indicating that **14** is the product of NvfI. To facilitate the isolation of **14**, we created another mutant in which a non-ribosomal peptide synthase gene named *e-anaPS* (Supplementary Fig. 7 and Supplementary Table 2) was deleted, since the gene is involved in the production of *epi*-aszonalenin C[8,19] co-eluting with **14** (Fig. 2c). Interpretation of the NMR spectra suggested that the structures of **14** and **15** contain an endoperoxide linkage between C-13 and C-2′ (Supplementary Figs. 8, 9). To prove the existence of the peroxide, we sought to reduce the peroxide by palladium-catalyzed hydrogenolysis[20]. The reduction of **15** provided the single product **16** (Fig. 3), and the spiro-lactone **16** must be derived from **17** with two hydroxyl groups at C-13 and C-2′, thus confirming the presence of the peroxide at these positions in **15**. Unfortunately, our attempt to reduce **14** generated a complex mixture of products, but considering that **14** is biosynthetically related to **15**, we concluded that **14** has the same peroxide linkage as **15**. The endoperoxides **14** and **15** were named fumigatonoids A and B, respectively.

The *nvfF*Δ and *nvfG*Δ mutants produced metabolites that appeared to be shunt products. Asnovolin G[6] (**18**), obtained from the *nvfF*Δ strain, possesses a six-membered spiro-lactone moiety, which appears to be generated by the translactonization after the occurrence of the hydroxyl group at C-13. Compound **19** isolated from the *nvfG*Δ strain, named novofumigatonol, was almost identical to **1**, and only differed by the hydroxyl group at C-5′. Since it is unlikely that **18** and **19** serve as the direct precursors of **1**, these compounds are likely synthesized in diverged pathways from that for **1**, as discussed later.

Taken together, our results indicate that the late-stage biosynthesis of novofumigatonin (**1**) is initiated by the oxidation of **2** to **14** by NvfI (Fig. 2a), which exhibits sequence similarity

with Fe(II)/αKG-dependent dioxygenases. The endoperoxide **14** would then be accepted by NvfD for the subsequent transformations. Since the methyl ester group present in **14** is missing in the metabolites obtained from the *nvfE*Δ, *nvfF*Δ, and *nvfG*Δ strains, NvfD is apparently involved in the lactonization process. To allow lactonization, the stereochemistry at C-5′ of **14** must be inverted. Although NvfD is predicted to be an α/β hydrolase, an example exists in which an α/β hydrolase also possesses an epimerase activity[21]. NvfD could therefore be the epimerase that converts **14** to its C-5′ epimer, which then undergoes spontaneous or NvfD-catalyzed lactonization to yield **20**. The following step utilizes the ketoreductase, NvfG, to produce **15**. Finally, NvfE and NvfF, which are both predicted to be Fe(II)/αKG-dependent dioxygenases, would convert **15** into the end product, novofumigatonin (**1**). We sought to obtain the predicted NvfD product **20** by heterologously expressing the ten genes according to the predicted pathway (Fig. 2e, lane iii), but unfortunately failed to isolate any *nvfD*-specific metabolites from the transformant, probably due to the instability of the compound. Similarly, our attempt to isolate the genuine product from the NvfE-catalyzed reaction was not successful, as the further addition of *nvfE* and *nvfG* to the above constructed transformant only generated **18**, which seems to be a shunt product, as an *nvfE*-specific metabolite (Fig. 2e, lane v). Nevertheless, the heterologous expression systems with 13 genes including *nvfF* produced **1** (Fig. 2e, lane vi and Fig. 2f), strongly indicating that the *nvf* cluster contains all of the genes required for the novofumigatonin synthesis.

**Characterization of NvfI as an endoperoxidase**. The gene-deletion experiments indicated involvement of endoperoxy precursors for orthoester generation, but the endoperoxidation mechanism remained to be clarified. To obtain deeper insight into the endoperoxide synthesis, the predicted Fe(II)/αKG-dependent dioxygenase NvfI was purified as a His6-tagged protein for in vitro enzymatic reactions (Supplementary Fig. 10).

Based on the predicted function, the recombinant NvfI protein was reacted with asnovolin A (**2**) under conditions similar to those used for other αKG-dependent enzymes in our previous studies[16,22]. Interestingly, the reaction did not provide the expected product **14**, but yielded its isomer **21** (Fig. 4a, lane iv). Compound **21** was isolated from a large-scale enzymatic reaction and determined to possess a hemiacetal linkage between C-13 and C-2′, instead of the endoperoxide of **14**, which seems to be derived from a rearrangement of the endoperoxide by the iron

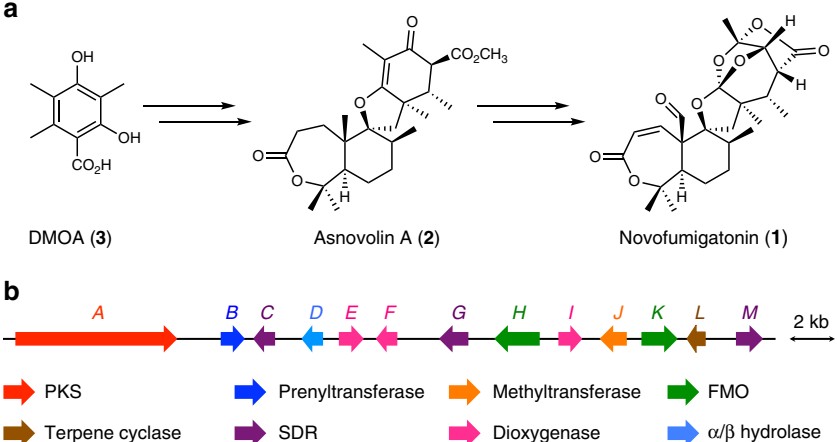

**Fig. 1** Novofumigatonin and its biosynthetic gene cluster. **a** Structures of novofumigatonin (**1**) and its predicted biosynthetic precursors. **b** Schematic representation of the *nvf* cluster and predicted function of each gene based on BLASTP comparison with characterized proteins. PKS polyketide synthase, FMO flavin-dependent monooxygenase, SDR short-chain dehydrogenase/reductase

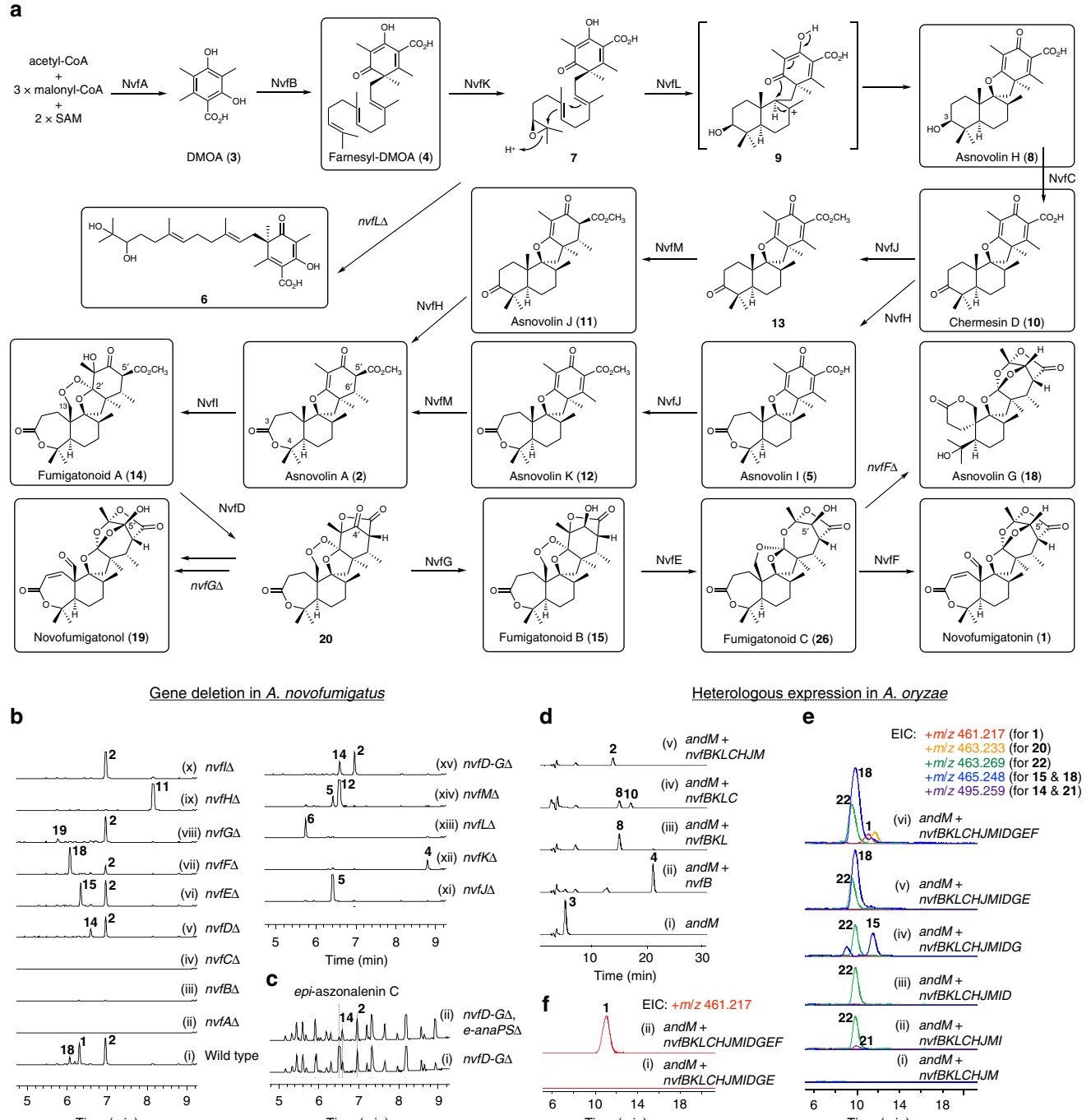

**Fig. 2** Biosynthesis of novofumigatonin. **a** Proposed biosynthetic pathway of novofumigatonin (**1**). Boxed compounds were isolated in this study. Detailed procedures for isolation and structural determination of the metabolites are provided in the Supplementary Notes 1 to 18. **b** LC–MS analysis of metabolites from gene-deletion mutants. Extracted ion chromatograms (EICs) at m/z 423.251 (for **4**), 431.243 (for **5**), 431.279 (for **11**), 445.259 (for **12**), 447.275 (for **2**), 457.256 (for **6**), 461.217 (for **1**), 465.248 (for **15** and **18**), 477.212 (for **19**), and 495.259 (for **14**), are shown. **c** LC–MS analysis after the deletion of e-anaPS. Base peak chromatograms are shown. **d** HPLC analysis of the metabolites from *A. oryzae* transformants. Chromatograms were monitored at 300 nm. **e**, **f** LC–MS analyses of the metabolites from *A. oryzae* transformants. The structures of compounds **21** and **22** are shown in Fig. 4

added to the reaction mixture (Fig. 4d). Therefore, we performed the reaction in the absence of exogenous iron, in which iron ions copurified with NvfI can be used for the catalysis. As a result, the reaction successfully generated **14** as the major product (Fig. 4a, lane v), demonstrating that NvfI is solely responsible for the endoperoxidation reaction. The reaction also yielded the mono-hydroxylated asnovolin B[6] (**22**) as a minor product, which was also detected from *A. oryzae* transformants harboring *nvfI*

(Figs. 2e, 4d). Furthermore, metal- and αKG-dependences of NvfI were confirmed by observations that the reaction was abolished in the presence of EDTA or in the absence of αKG (Fig. 4a, lanes ii and iii).

As an αKG-dependent enzyme, the catalysis by NvfI should follow the common mechanism seen in this class of oxidative enzymes[23]: an Fe(IV)-oxo species is generated by the oxidative decarboxylation of αKG to abstract a hydrogen atom from **2** to

**Fig. 3** Palladium-catalyzed hydrogenolysis of **15**. Reductive cleavage of the endoperoxide proved the presence of the endoperoxide linkage in **15**

initiate the reaction. Thus, the mechanism for the endoperoxidation by NvfI is proposed as follows (Fig. 4e): the hydrogen atom at C-13 is abstracted to give the radical species **23**; a molecular oxygen is then incorporated to generate the peroxy radical **24**, which undergoes C-O bond formation at C-2′ to yield **25**; and an oxygen rebound at C-3′ completes the reaction to provide **14**. This proposed mechanism for the endoperoxidation is similar to that proposed for the fumitremorgin B endoperoxidase (FtmOx1)[24,25], which is another αKG-dependent enzyme. However, the mechanisms for the radical quenching differ in these two enzymes; FtmOx1 requires a reducing agent as a hydrogen donor at the end of the reaction, while NvfI completes the reaction by the oxygen rebound, altogether introducing three oxygen atoms to the substrate. For the formation of the minor product **22**, the hydrogen abstraction would occur on the C-7′ methyl group, which is immediately followed by the rebound of the hydroxyl radical (Fig. 4d).

**Mechanistic investigation of the orthoesterification**. To unveil how the endoperoxy precursors are transformed into novofumigatonin (**1**) with its unique orthoester moiety, two other putative Fe(II)/αKG-dependent dioxygenases, NvfE and NvfF, were also purified as His$_6$-tagged proteins (Supplementary Fig. 10). Initially, the reaction by NvfE was investigated using similar reaction conditions to those for NvfI. The reaction with **15** as a substrate gave two isomeric products (Fig. 4b, lane v), one of which was identified as **18** obtained from the *A. novofumigatus nvfF*Δ strain. The other product **26** was isolated from a large-scale reaction for the structural characterization, and quite surprisingly, it was elucidated that **26** has a different orthoester moiety from those seen in **1** and **18** (Figs. 2a, 4f), in which the hydroxyl group at C-5′ is not involved in the orthoester formation. We also found that **26** is easily transformed into **18** when formic acid was added after the completion of the reaction (Fig. 4b, lane vi). Thus, we reasoned that **18** is not an enzymatic product, and that **26** undergoes an acid-catalyzed rearrangement to yield **18** (Supplementary Fig. 11).

To evaluate the intermediacies of **18** and **26** in the novofumigatonin pathway, these compounds were individually incubated with NvfF, which seems to be responsible for the very last steps of the biosynthesis. As a result, **26** was efficiently converted to **1** in a metal- and αKG-dependent manner (Fig. 4c, lanes iii to vi), but **18** was not utilized as a substrate of NvfF (Fig. 4c, lanes i and ii), indicating that **26** is the genuine product and substrate of NvfE and NvfF, respectively, and that **18** is a shunt pathway product. Collectively, NvfF catalyzes two rounds of oxidation to transform **26** into the end product, novofumigatonin (**1**).

Despite the sequence similarity between NvfE and the known Fe(II)/αKG-dependent dioxygenases, the reaction catalyzed by NvfE is not an oxidation, but an isomerization event, which led us to further investigate this intriguing enzyme. Surprisingly, the reaction was not inhibited in the absence of αKG or in the presence of 20 mM EDTA, and only the purified enzyme and substrate were required for the complete reaction (Fig. 4b, lanes ii to iv). This observation suggested that NvfE is a cofactor/cosubstrate-free enzyme. We thus carefully analyzed the sequence alignment between NvfE and other αKG-dependent enzymes involved in fungal secondary metabolism (Supplementary Fig. 12). As a result, we found that one highly conserved glutamine residue important for the αKG binding[24,26–28] is substituted with glutamate (E149) in NvfE, while all of the residues consisting of the iron-binding facial triad[29] (H152, D154, and H234) are still conserved. To reveal the importance of these residues, four genes containing each of the following codon changes, E149Q, H152A, D154A, and H234A, were constructed and the resulting mutant enzymes were subjected to in vitro enzymatic reactions. When incubated only with substrate **15**, each of the mutant enzymes exhibited reduced activity but did not completely lose activity (Supplementary Fig. 13). Quite interestingly, the reactions by all four of the mutants were enhanced in the presence of $Fe^{2+}$ and/or ascorbate and inhibited in the presence of EDTA (Supplementary Fig. 13). Altogether, these results show that the orthoester formation proceeds in a metal-dependent manner. The differences between the reactions catalyzed by wild-type NvfE and the mutants could be attributed to the fact that E149 participates in the iron binding in the active site via its carboxyl group together with the other three residues, resulting in stronger iron-binding affinity. However, this hypothesis should be confirmed by a future X-ray crystallographic study on NvfE complexed with iron.

It is well known that Fe(II) causes the cleavage and rearrangement of endoperoxide linkages in a variety of compounds, such as the antimalarial agent artemisinin[30]. Thus, we reasoned that the NvfE-catalyzed orthoesterification is also initiated by the iron-mediated cleavage of the endoperoxide and proposed the following reaction mechanism (Fig. 4f). First, the ferrous iron in the active site would bind to the oxygen atom at C-13, to cleave the endoperoxide. The oxygen-centered radical **27** generated herein then undergoes β-scission to give the carbon-centered radical **28**, which is oxidized by Fe(III) to form the carbocationic species **29**. Finally, the carbocation is quenched by the two rounds of heterocyclization, to afford the orthoester **26**.

The details of the reactions catalyzed by NvfF to finalize the biosynthesis are still not very clear, as we were not able to obtain the product from the first-round reaction. Nevertheless, it is evident that NvfF is engaged in two sequential oxidative reactions, the dehydrogenation to introduce the C-C double bond between C-1 and C-2 and the aldehyde formation at C-13. Homologs of NvfF are often found in the DMOA-derived meroterpenoid pathways, and most of them catalyze multistep reactions as NvfF[16,22,31,32]. Interestingly, contrary to the commonly seen desaturation event, the aldehyde forming reaction by NvfF is relatively rare for reactions catalyzed by αKG-dependent enzymes. Based on the reaction mechanism proposed for aldehyde formation by this class of dioxygenases[33,34], the mechanism for the NvfF-catalyzed reaction to yield the aldehyde with the rearranged orthoester could be

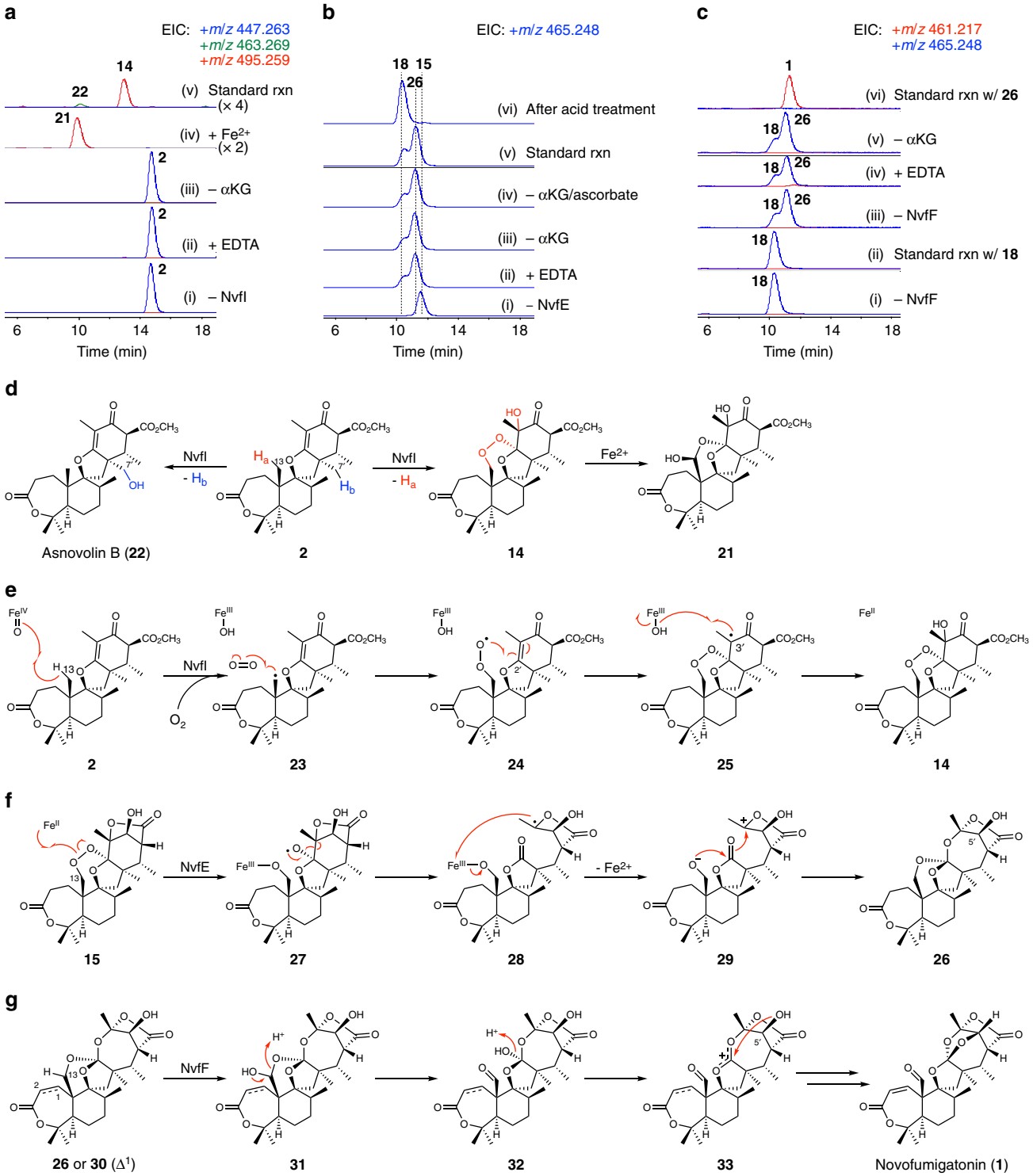

**Fig. 4** Characterization of the key enzymes for novofumigatonin orthoesterification. **a–c** LC–MS analyses of the products from enzymatic reactions of **a** NvfI and **2**, **b** NvfE and **15**, and **c** NvfF and **18** (lanes i and ii) or **26** (lanes iii to vi). **d** Reactions catalyzed by NvfI and the Fe(II)-catalyzed rearrangement of **14**. NvfI produces **14** as the major product and **22** as a minor product. **e** Proposed reaction mechanism for the endoperoxidation by NvfI. **f** Proposed mechanism for the orthoesterification by NvfE. **g** Proposed mechanism for the aldehyde formation and transorthoesterification of the predicted NvfF product

proposed as follows (Fig. 4g). Initially, the ferryl-oxo species generated in the active site of the enzyme abstracts a hydrogen atom from C-13 of **26**, or its dehydrogenated form **30**, and then hydroxylates this position, which would be the only role of NvfF in this transformation. The resultant hemiacetal **31** is sponta-neously converted to an aldehyde, by which a hemiorthoester group is generated as well. The resulting hemiorthoester **32** would then be transformed to the orthoester via the cationic intermediate **33** by the participation of the hydroxyl group at C-5′, thus completing the reaction. These proposed mechanisms also explain the formation of **19** in the absence of the ketoreductase gene *nvfG* (Supplementary Fig. 11). In this case,

**Fig. 5** Biosynthesis of fumigatonin. Structure of fumigatonin and reactions required to synthesize fumigatonin from **26** are shown

**20**, the 4′-keto analog of **15**, would be accepted by NvfE and NvfF to undergo similar rearrangement and oxidation reactions as shown in Fig. 4f, g but a water molecule is used for the neutralization at the last step of the reaction to give the orthoester of **19**.

## Discussion

In this study, we elucidated the biosynthetic pathway of novofumigatonin (**1**) by CRISPR-Cas9-guided gene-deletion experiments, heterologous fungal expression, and in vitro enzymatic reactions. We believe that we have demonstrated the powerfulness of combining these three different approaches to efficiently elucidate a fungal natural product pathway and to identify key enzymes for the biosynthesis.

We have identified and characterized the endoperoxidase NvfI and the endoperoxide isomerase NvfE for the orthoester synthesis. The reactions catalyzed by NvfI and NvfE somewhat resemble those by prostaglandin endoperoxide H synthases (PGHSs) and prostacyclin or thromboxane A synthases, respectively[35], but differ in that these enzymes involved in prostanoid biogenesis are heme proteins, while NvfI and NvfE utilize non-heme iron to perform a similar set of reactions. Enzymatic endoperoxidation has been well studied in PGHSs[35], but except for the PGHSs, fumitremorgin B endoperoxidase[24] is the only known enzyme that synthesizes an endoperoxide as a product, despite the wide occurrence of peroxy natural products[36]. Interestingly, NvfI shares little amino acid sequence similarity with the known endoperoxidases, and NvfI thus represents a rare example of an endoperoxide-producing enzyme.

It is remarkable that the orthoester synthesis is achieved by a non-heme iron-dependent isomerase, NvfE. Non-heme iron-utilizing enzymes are widespread in nature and perform versatile reactions[37], but are mostly involved in oxidative reactions. Enzymatic orthoesterification can be performed by the non-heme iron-dependent enzymes for the orthosomycins pathways as well[4], but their orthoester synthesis is an oxidative reaction without any structural rearrangement, thus completely differing from the NvfE-catalyzed reaction. Nevertheless, there are some examples in which isomerization reactions are catalyzed by NvfE homologs, such as CarC[38], SnoN[39], and AndA[16], in the carbapenem, nogalamycin, and anditomin pathways, respectively. The reactions by these enzymes, however, are all αKG-dependent, and they retain their functions as oxidative enzymes as well. In contrast, NvfE appears to have completely lost the dioxygenase activity, and to the best of our knowledge, NvfE is the first reported enzyme that gained the isomerase activity by losing the αKG-binding ability.

Other intriguing findings regarding the NvfE-catalyzed reaction are that the resultant orthoester is different from that of **1** and that the NvfE product **26** completely possesses the same backbone structure as that of another orthoester-containing natural product, fumigatonin[40] (Fig. 5). We previously speculated that the two different orthoesters in **1** and fumigatonin are respectively biosynthesized in somewhat different manners[7], but our present study revealed that the novofumigatonin-type orthoester actually originates from the fumigatonin-type orthoester. Overall, **26** would be the common precursor for the biosyntheses of both **1** and fumigatonin, and the biosynthetic gene cluster for fumigatonin should contain a few more genes as compared with those of the *nvf* cluster, to allow for the C-6 hydroxylation and the two acetylations (Fig. 5).

In conclusion, we have revealed the complex biosynthetic route to novofumigatonin (**1**) and discovered the intriguing endoperoxide isomerase, NvfE, for the orthoester formation, which was accomplished by establishing genetic engineering in *A. novofumigatus*. Further biochemical and biophysical characterizations of NvfE will clarify this unusual chemistry and could even provide opportunities to engineer known αKG-dependent enzymes into αKG-independent biocatalysts to expand nature's catalytic versatility.

## Methods

**General**. Solvents and chemicals were purchased from Sigma-Aldrich, VWR International, Fisher Scientific International, Inc., Wako Chemicals Ltd., or Kanto Chemical Co., Inc., unless noted otherwise. Oligonucleotide primers (Supplementary Data 1) were purchased from Integrated DNA Technologies Inc., Eurofins Genetics, or Sigma-Aldrich. PCR was performed using a T100™ Thermal Cycler (BIO-RAD) or a TaKaRa PCR Thermal Cycler Dice® Gradient (TaKaRa), with PfuX7 DNA polymerase[41], iProof DNA polymerase (BIO-RAD), or Phire Plant Direct PCR Master Mix (Thermo Scientific). Sequence analyses were performed by Eurofins Genetics. Flash chromatography was performed using an Isolera flash purification system (Biotage). Preparative HPLC was performed using a Waters 600 controller with a 996 photodiode array detector (Waters) or a Shimadzu Prominence LC-20AD system. NMR spectra were obtained with Bruker AVANCE 400 MHz, 600 MHz, or 800 MHz spectrometers at DTU NMR Center • DTU, or JEOL ECX-500 or ECZ-500 spectrometers, and chemical shifts were recorded with reference to solvent signals ($^1$H NMR: CDCl$_3$ 7.26 ppm, CD$_3$OD 3.34 ppm, DMSO-$d_6$ 2.49 ppm, acetone-$d_6$ 2.05 ppm; $^{13}$C NMR: CDCl$_3$ 77 ppm, CD$_3$OD 49 ppm, DMSO-$d_6$ 39.5 ppm, acetone-$d_6$ 29.9 ppm). LC–MS samples from the gene-deletion experiments were injected into a Dionex Ultimate 3000 UHPLC system (Thermo Scientific)—a maXis 3G QTOF orthogonal mass spectrometer (Bruker Daltonics), using electrospray ionization with a Kinetex C$_{18}$ column (2.1 i.d. × 100 mm; Phenomenex). LC–MS samples from the heterologous expression and in vitro experiments were injected into a Shimadzu Prominence LC-20AD system with a compact mass spectrometer (Bruker Daltonics), using electrospray ionization with a COSMOSIL 2.5C$_{18}$-MS-II column (2 i.d. × 75 mm; Nacalai Tesque, Inc.).

**Strains**. *Aspergillus novofumigatus* IBT 16806 (IFM 55215) was obtained from the IBT Culture Collection at the Department of Biotechnology and Biomedicine, Technical University of Denmark, Denmark, and from the Medical Mycology Research Center, Chiba University, Japan. *Aspergillus oryzae* NSAR1 (niaD$^-$, sC$^-$, ΔargB, adeA$^-$)[17] was used as the host for fungal heterologous expression. Standard DNA engineering experiments were performed using *Escherichia coli* DH5α. *E. coli* Rosetta™2(DE3)pLysS (Novagen) was used for the expression of the NvfI, NvfE, NvfF, and NvfE mutants.

**Construction of plasmids for fungal transformations**. For the construction of plasmids for the CRISPR-Cas9 system, two fragments for the expression cassette of the targeted gene-specific single guide RNA (sgRNA) were amplified by PCR with pFC334 as the template, and ligated into the pFC332 vector[10], which had been digested with PacI and subsequently treated with Nt.BbvCI, by the Uracil-Specific Excision Reagent (USER) fusion method[42]. For the construction of plasmids for gene-deletion experiments, approximately 1-kb of the 5′ and 3′ flanking regions of the targeted gene were amplified by PCR from the genomic DNA of *A. novofumigatus* IBT 16806, and introduced into the PacI/Nt.BbvCI USER cassette sites of pU2002c[10]. The plasmids constructed in this study and the primers used for the construction of each plasmid are listed in Supplementary Data 2.

For the construction of fungal expression plasmids for *A. oryzae*, each gene in the *nvf* cluster, except for *nvfK* and *nvfI*, was amplified from *A. novofumigatus* IBT 16806 genomic DNA using the published genome sequence[8], with the primers listed in Supplementary Data 1 and 2, and initially ligated into the pTAex3 vector[43], while *nvfK* and *nvfI* were introduced into the pUSA[44] and pUNA[45] vectors, respectively, using an In-Fusion® HD Cloning Kit (TaKaRa Clontech). To construct multigene-containing vectors, fragments containing the *amyB* promoter (P*amyB*) and the *amyB* terminator (T*amyB*) were amplified from the pTAex3-based plasmids, and ligated into the previously constructed single gene-containing

vectors or other vectors, pAdeA[46], pPTRI[47], or pBARI[16]. The plasmids constructed in this study and the primers used for the construction of each plasmid are listed in Supplementary Data 2.

**Transformation of *A. novofumigatus* and *A. oryzae*.** To transform *A. novofumigatus* IBT 16806 and its mutants, they were first cultivated in 100 mL of minimum media (6 g L$^{-1}$ NaNO$_3$, 0.52 g L$^{-1}$ KCl, 0.52 g L$^{-1}$ MgSO$_4$·7H$_2$O, 1.52 g L$^{-1}$ KH$_2$PO$_4$, 10 g L$^{-1}$ D-glucose, 10 mg L$^{-1}$ thiamine, supplemented with 1 mL L$^{-1}$ of a trace element solution[48]) containing 1.12 g L$^{-1}$ uracil and 2.44 g L$^{-1}$ uridine, if necessary, for one to two days at 30 °C and at 160 rpm. Mycelia were then collected and incubated in TF solution 1 (40 mg mL$^{-1}$ Glucanex enzymes (Novozymes A/S), 0.6 M (NH$_4$)$_2$SO$_4$, 50 mM maleic acid, pH 5.5) for 3 h at 30 °C and at 160 rpm. The resultant protoplast-containing mixture was filtrated, and the filtrate was washed by TF solution 2 (10 mM Tris-HCl, pH 7.5, 1.2 M sorbitol, 50 mM CaCl$_2$, 35 mM NaCl). The protoplast was resuspended in TF solution 2 to a concentration of 1–5 × 10$^7$ cells mL$^{-1}$ and mixed with the transformation plasmid(s), which was further incubated for 30 min at room temperature. A total of 1.35 mL of TF solution 3 (10 mM Tris-HCl, pH 7.5, 60 % PEG4000, 50 mM CaCl$_2$) was then added in three portions to the mixture of the protoplast and plasmids. After 20 min incubation at room temperature, the mixture was diluted with 5 mL of TF solution 2 and 6 mL of the molten transformation agar media (6 g L$^{-1}$ NaNO$_3$, 0.52 g L$^{-1}$ KCl, 0.52 g L$^{-1}$ MgSO$_4$·7H$_2$O, 1.52 g L$^{-1}$ KH$_2$PO$_4$, 10 g L$^{-1}$ D-glucose, 188.2 g L$^{-1}$ sorbitol, 10 mg L$^{-1}$ thiamine, and 20 g L$^{-1}$ agar supplemented with 1 mL L$^{-1}$ of a trace element solution[48]), which was poured on the top of the pre-made transformation agar media plate. The plate was further incubated for about a week at 25 °C until mutant colonies appear.

For the deactivation of the *pyrG* gene (Protein ID in JGI database: 399623), pFC332-Aspnov_pyrG was used for transformation of *A. novofumigatus* IBT 16806. Transformants were selected on transformation media supplemented with 1.12 g L$^{-1}$ uracil, 2.44 g L$^{-1}$ uridine, and 100 mg L$^{-1}$ hygromycin B. Mutant strains with the *pyrG$^-$* genotype were selected on minimum agar media (6 g L$^{-1}$ NaNO$_3$, 0.52 g L$^{-1}$ KCl, 0.52 g L$^{-1}$ MgSO$_4$·7H$_2$O, 1.52 g L$^{-1}$ KH$_2$PO$_4$, 10 g L$^{-1}$ D-glucose, 10 mg L$^{-1}$ thiamine, 20 g L$^{-1}$ agar, supplemented with 1 mL L$^{-1}$ of a trace element solution[48]), containing 1.30 g L$^{-1}$ of 5-fluoroorotic acid (5-FOA) to obtain transformants in which the *pyrG* gene is deactivated, since the wild-type strain is sensitive to 5-FOA. The selected *pyrG$^-$* strain was then transformed to delete the *ligD* gene (Protein ID in JGI database: 439258), using both pFC332-Aspnov_ligD and pU2002c-Aspnov_ligD, and the transformant was selected on transformation media supplemented with hygromycin B. Transformants were then cultivated on minimal media supplemented with uracil and uridine, to induce the elimination of the *pyrG* marker gene (*AfpyrG*: *pyrG* from *Aspergillus flavus*) via intramolecular homologous recombination (Supplementary Fig. 3), and were further selected with 5-FOA, hereby creating the *pyrG$^-$*, *ligD*Δ strain.

The *pyrG$^-$*, *ligD*Δ strain was transformed with the pU2002c-based plasmids corresponding to the targeted gene(s) (Supplementary Data 2), and transformants were selected on minimal media. When deleting *nvfK*, *nvfM*, and *nvfD-G*, pFC332-based plasmids (Supplementary Data 2) were used as well, since transformations with only pU2002c-based plasmids were inefficient. To construct the strain lacking the *nvfD-G* and *e-anaPS* genes, the *AfpyrG* gene was eliminated from the *pyrG$^-$*, *ligD*Δ, *nvfD-G*Δ::*AfpyrG* strain, and the resulting mutant was transformed with pFC332-e-anaPS and pU2002c-e-anaPS. The successful deletion of targeted genes was confirmed by colony-direct PCR of each transformant (Supplementary Fig. 4). The mutant strains of *A. novofumigatus* constructed in this study were deposited in the IBT Culture Collection at the Department of Biotechnology and Biomedicine, Technical University of Denmark (Supplementary Table 3).

To transform *A. oryzae* NSAR1 and its mutants, they were first cultivated in DPY medium (2% dextrin, 1% hipolypepton (Nihon Pharmaceutical Co., Ltd.), 0.5% yeast extract (Difco), 0.5% KH$_2$PO$_4$, and 0.05% MgSO$_4$·7H$_2$O) for one to two days at 30 °C and at 160 rpm. The general transformation procedure is the same as described for *A. novofumigatus* except that Yatalase (TaKaRa) was used as fungal cell lytic enzymes at a concentration of 10 mg mL$^{-1}$. Transformants were selected on M-sorbitol media (0.2% NH$_4$Cl, 0.1% (NH$_4$)$_2$SO$_4$, 0.05% KCl, 0.05% NaCl, 0.1% KH$_2$PO$_4$, 0.05% MgSO$_4$·7H$_2$O, 0.002% FeSO$_4$·7H$_2$O, 2% glucose, and 1.2 M sorbitol, pH 5.5) or CD-sorbitol media (0.3% NaNO$_3$, 0.2% KCl, 0.1% KH$_2$PO$_4$, 0.05% MgSO$_4$·7H$_2$O, 0.002% FeSO$_4$·7H$_2$O, 2% glucose, and 1.2 M sorbitol, pH 5.5) supplemented with 0.1% arginine, 0.15% methionine, 0.05% adenine, 0.1 μg mL$^{-1}$ pyrithiamine hydrobromide, and/or 50 μL mL$^{-1}$ glufosinate solution[16] depending on the selection markers used in the transformation. The plates were further incubated for about a week at 30 °C until mutant colonies appear. Transformants of *A. oryzae* constructed in this study and plasmids used for each transformation are listed in Supplementary Table 4.

**LC-MS analyses.** *Aspergillus novofumigatus* IBT 16806 and its mutants were cultivated on YES agar plates (20 g L$^{-1}$ yeast extract, 150 g L$^{-1}$ sucrose, 0.5 g L$^{-1}$ MgSO$_4$·7H$_2$O, 20 g L$^{-1}$ agar supplemented with 1 mL L$^{-1}$ of a trace element solution, pH 6.5) at 25 °C for seven days, whereafter metabolites were extracted with ethyl acetate using ultrasonication. Separation was performed with a solvent system of water containing 20 mM formic acid (solvent A) and acetonitrile containing 20 mM formic acid (solvent B), at a flow rate of 0.4 mL min$^{-1}$ and a column temperature of 40 °C, using the following program: a linear gradient from

10:90 (solvent B/solvent A) to 100:0 for 10 min, 100:0 for the following 3 min, and a linear gradient from 100:0 to 10:90 within the following 2 min.

Transformants of *A. oryzae* were grown in shaking cultures in DPY medium for three days at 30 °C and at 160 rpm, and the culture broth was extracted with ethyl acetate. Separation was performed with a solvent system of water containing 20 mM formic acid (solvent A) and acetonitrile containing 20 mM formic acid (solvent B), at a flow rate of 0.1 mL/min and a column temperature of 40 °C, using the following program: 50:50 (solvent B/solvent A) for 3 min, a linear gradient from 50:50 to 100:0 within the following 17 min, 100:0 for 3 additional min, and a linear gradient from 100:0 to 50:50 within the following 1 min.

Analytical conditions for products from in vitro enzymatic reactions were the same as those described for metabolites from *A. oryzae* transformants.

**Isolation and purification of each metabolite.** For the isolation of each metabolite from *A. novofumigatus*, wild type and mutants were inoculated on 100 to 200 YES agar plates (ca. 2–4 L), respectively, and cultivated for ~10 days at 25 °C. The resulting fungal cultures were extracted with ethyl acetate twice, fractionated by flash chromatography, and purified by preparative HPLC. For the isolation of each metabolite from *A. oryzae* transformants, media from 1 to 4 L of the culture were extracted with ethyl acetate. Mycelia were extracted with acetone at room temperature, concentrated, and reextracted with ethyl acetate. Both extracts were combined and subjected to open column chromatography and further purification by preparative HPLC. The detailed purification procedures and structural characterizations of the compounds are described in Supplementary Notes 1 to 18, and the spectral data are provided in Supplementary Figs. 14 to 100 and Supplementary Tables 5 to 22.

**Hydrogenolysis of fumigatonoid B (15).** To a stirred solution of **15** (4.4 mg, 9 μmol) in isopropanol (14 mL) was added 10% palladium on activated charcoal (~5 mg), and the mixture was stirred for 5 h under a hydrogen atmosphere. The suspension was then filtered through a PTFE syringe filter (0.45 μm), and the solvent was removed in vacuo. The crude product was purified by reverse-phase preparative HPLC (40% aqueous acetonitrile, 5 mL min$^{-1}$) on a Luna II C18 column (250 × 10 mm, 5 μm, Phenomenex), to yield 3.6 mg (82%, 7.5 μmol) of a white amorphous solid. HR-ESI-MS found *m/z* 489.2463 [M+Na]$^+$ (calcd. 489.2459 for C$_{25}$H$_{38}$O$_8$Na).

**Expression and purification of the Nvf enzymes.** To express *nvfI*, *nvfE*, and *nvfF* in *E. coli*, complementary DNA (cDNA) for each gene was introduced into the pET-28a(+) vector (Novagen), using an In-Fusion® HD Cloning Kit (Supplementary Data 2). To obtain an intron-free *nvfI* gene, total RNA was extracted from an *A. oryzae* transformant expressing *nvfI* using ISOGEN (Nippon Gene Co., Ltd.), and cDNA was synthesized with SuperScript™ III Reverse Transcriptase (Invitrogen) from the extracted RNA. The intron-free *nvfE* gene was generated by amplifying and ligating the two predicted exons. The *nvfF* gene has no intron, and therefore it was directly amplified from *A. novofumigatus* genomic DNA. NvfE mutants were constructed by PCR methods, using pairs of the mutation primers.

For the expression of NvfI, NvfE, NvfF, and NvfE mutants, *E. coli* Rosetta™2 (DE3)pLysS was transformed with the pET-28a(+)-based plasmid for each gene. The transformants were incubated with shaking at 37 °C/160 rpm, in ZYM-G medium[49] supplemented with 50 mg L$^{-1}$ kanamycin sulfate and 12.5 mg L$^{-1}$ chloramphenicol. When the cultures had grown to an OD600 of 0.6, gene expression was induced by the addition of 0.5 mM IPTG. Then, the incubation was continued for 14 h at 30 °C/160 rpm. The cells were harvested by centrifugation and resuspended in lysis buffer (50 mM Tris-HCl, pH 7.5, 150 mM NaCl, 5 mM imidazole, 5% glycerol). After lysis on ice by sonication, the cell debris was removed by centrifugation. The supernatant was loaded onto a Ni-NTA affinity column. Unbound proteins were removed with 30 column volumes of wash buffer (50 mM Tris-HCl, pH 7.5, 150 mM NaCl, 10 mM imidazole, 5% glycerol). His-tagged proteins were eluted with 5 column volumes of elution buffer (50 mM Tris-HCl, pH 7.5, 150 mM NaCl, 300 mM imidazole, 5% glycerol). The purity of the enzymes was analyzed by sodium dodecyl sulfate polyacrylamide gel electrophoresis (SDS-PAGE) (Supplementary Fig. 10). The protein concentrations were determined using a SimpliNano Spectrophotometer (GE Healthcare Life Sciences).

**Enzymatic reaction assay of purified proteins.** All of the enzymatic reactions were performed in 50 μL reaction mixtures at 30 °C, and terminated by adding 50 μL of methanol and vortex mixing, and then the supernatant obtained after centrifugation was analyzed by LC–MS.

The standard enzymatic reaction of NvfI with asnovolin A (**2**) was performed in reaction mixtures containing 50 mM Tris-HCl buffer (pH 7.5), 250 μM of **2**, 2.5 mM α-ketoglutarate, 4 mM ascorbate, and 9.6 μM NvfI, for 2.5 h. When necessary, 0.1 mM FeSO$_4$ or 1 mM EDTA was added to the reaction.

The standard enzymatic reaction of NvfE with fumigatonoid B (**15**) was performed in reaction mixtures containing 50 mM Tris-HCl buffer (pH 7.5), 50 μM of **15**, 2.5 mM α-ketoglutarate, 4 mM ascorbate, and 20.7 μM NvfE, for 2.5 h. When required, 20 mM EDTA was added to the reaction. To investigate the stability of **26**, formic acid (final conc. 0.5%) was added after the completion of the reaction.

The standard enzymatic reaction of NvfF with asnovolin G (**18**) or fumigatonoid C (**26**) was performed in reaction mixtures containing 50 mM Tris-HCl buffer (pH 7.5), 50 μM of **18** or **26**, 2.5 mM α-ketoglutarate, 4 mM ascorbate, and 11.1 μM NvfF, for 1 h. When required, 1 mM EDTA was added to the reaction.

The standard enzymatic reaction of NvfE or its mutants with fumigatonoid B (**15**) was performed in reaction mixtures containing 50 mM Tris-HCl buffer (pH 7.5), 50 μM of **15**, and 4 μM NvfE or its mutants, for 1.5 h. When required, 0.1 mM $FeSO_4$, 4 mM ascorbate, or 1 mM EDTA was added to the reaction.

**Data availability**. The authors declare that the data supporting the findings of this study are available within the article and its Supplementary Information and from the corresponding authors upon reasonable request.

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

## Acknowledgements

The NMR Center • DTU, and the Villum Foundation are acknowledged for access to the 400, 600, and 800 MHz NMR spectrometers. We thank Dr. Kasper Enemark-Rasmussen for his assistance on acquiring NMR data. We are grateful to Prof. Katsuya Gomi (Tohoku University) and Prof. Katsuhiko Kitamoto (The University of Tokyo) for kindly

providing the expression vectors and the fungal strain. This work was in part supported by postdoctoral fellowship from the Novo Nordisk Foundation to Y.M. (NNF15OC0015172), by a Villum Foundation Young Investigators Programme grant (VKR023427) to M.R.A., and by a Grant-in-Aid for Scientific Research from the Ministry of Education, Culture, Sports, Science and Technology, Japan (JSPS KAKENHI Grant Number JP15H01836 and JP16H06443) to I.A.

## Author contributions

Y.M., I.A., and T.O.L designed the research. Y.M. performed the gene-deletion experiments, heterologous fungal expression, in vitro enzymatic reactions, and structural characterization of new compounds. T.B. constructed the heterologous fungal expression systems. C.B.W.P. synthetically derivatized compound **15** for the structural characterization. C.S.N. and U.H.M. developed the methodology for the gene-deletion experiments in *A. novofumigatus*. Y.M., I.K., T.C.V., and M.R.A. discovered and analyzed the novofumigatonin biosynthetic gene cluster. Y.M., C.H.G., I.A., and T.O.L. analyzed the data. Y.M., C.B.W.P., I.A., and T.O.L. wrote the paper. All the authors reviewed the manuscript.

## Additional information

**Competing interests:** The authors declare no competing interests.

