## [Peer Review File · Nature Communications]

Reviewers' comments:

Reviewer #1 (Remarks to the Author):

The manuscript describes the elucidation of the biosynthetic pathway of the fungal natural product novofumigatonin using a combination of in vivo gene inactivations, heterologous production, and in vitro characterization of 3 enzymes. Several metabolites were identified and structurally elucidated from the genetic work, which helped lay out a clear biosynthetic scheme that is shown in Figure 2. The most interesting aspect of the manuscript is the discovery of a peroxide forming dioxygenase and an orthoester forming, iron-dependent enzyme. The experimental design is rigorous and a very impressive amount of experimental results are provided. The conclusions are strongly supported by the data. Overall this is an excellent contribution. The only considerations I have are:

1. In some regard, the manuscript is too dense as a significant portion is dedicated to results from gene inactivations of early- to middle-stage reactions, which are more so expected than the late-stage reactions. It isn't until page 10 when the endoperoxide chemistry (formation and utilization to form the orthoester) is finally introduced. It took a considerable amount of time to work through the significance of the gene inactivation experiments in relationship to what is known for other related terpenoids. I found this somewhat distracting since the interesting part is the endoperoxide and orthoester forming chemistry.

My advice would be to separate into two separate manuscripts or somehow streamline the early- and middle-stage results.

2. In the first paragraph of the discussion, it is stated that their approach enabled them to "rapidly elucidate". I am not sure about the use of rapidly?

3. Noticed that Supplemental Figure 2 is not cited in the main text until after the introduction of several other Supplemental figures. These figures might need to be renumbered.

4. In the final, conclusion paragraph, the meaning of the statement "would even expand the catalytic versatility of the known aKG-dependent enzymes" is unclear. Potentially reword

Reviewer #2 (Remarks to the Author):

The manuscript by Matsuda et al. described the experimental evidences for biosynthetic pathway of meroterpenoid novofumigatonin by a number of gene knockouts, heterologous gene expression in *Aspergillus oryzae* and in vitro analysis of key enzymes. Early-stage transformations to asnobolin A via farnesyl-DMOA and asnobolin H are rather standard because the similar transformations are found in biosynthesis of related meroterpenoids such as andrastin, terretonin and austinol. However, a series of oxidative transformations from asnobolin A to novofumigatonin are highly intriguing especially in view of involvement of endoperoxide 14 and its conversion to orthoester by endoperoxide isomerization. These enzymes NvfI and NvfE catalyzed the reactions that are rarely found in the literatures. This reviewer recognizes that the data presented are clear and does give interests to the readership of Nature Communication. This reviewer recommends publication following minor revisions.

1) Most of the researchers are interested in CRISPER-Cas9 based genome editing. The authors did not use this technique although they can be applied to all gene deletion experiments. This reviewer suggests to add some comments to explain the reason for this.

2) Page 10, line 9, "hydrogenation" should be "hydrogenolysis".

3) In Figure 4, as NvfF in Figure 4g, addition of NvfI and NvfE in Figures 4e and 4f may help to differentiate these similar enzymatic reactions.

4) In Figure 5, "acetoxylation" should be "hydroxylation-acetylation".

5) Page 11, line 5 from the bottom, "the stereochemistry of the methyl ester group ---" is not correct. It should be changed to "the stereochemistry at C5' ---".

6) Page 12, line 5-7, the meaning of the sentence "Similarly, our attempt ----- metabolite (Fig. 2e, lane v)" is hard to understand. The phrase "the other eleven genes" should be "the other twelve

genes". Please check it.

Reviewer #3 (Remarks to the Author):

Overview:

This study by Matsuda et al describes the biosynthetic pathway of a structurally elaborate meroterpoid derived terpene novofumigatonin. The boundaries of the biosynthetic gene cluster were unambiguously determined via heterologous expression, and a proposed sequence of biochemical transformations resulting in novofumigatonin was based on a series of overlapping CRISPR-generated targeted gene disruptions, heterologous expression experiments, and in vitro turnover assays. In the process several novel transformations were uncovered including a rare endoperoxide synthase, an endoperoxide isomerase yielding an orthoester, and a orthoester isomerase interconverting orthoester constitutional isomers. The orthoester isomerase connects the biosynthesis of fumigatonin and novofumigatonin and highlights an interesting extension of the former.

The manuscript is meticulously prepared, experiments are well described, and results are statistically valid. Conclusions are drawn from experiments with adequate controls and orthogonal methods were used to validate most conclusions.

Critiques:

Regarding the mechanism of the endoperoxide forming enzyme NvfI, "a molecular oxygen is then incorporated to generate the peroxy radical 24, which undergoes C-O bond formation at C-2' to yield 25": There was a lack of discussion alternative mechanistic possibilities. The proposed generation of the C-13 radical is consistent with the canon, but the interception of this highly unstable species by an additional molecular oxygen to form a hydroperoxide is without precedent. Please provide precedent for the active site hosting two molecules of oxygen during a single reaction coordinate to generate a hydroperoxide.

Conclusions of NvfE mutagenesis seem overinterpreted. Since mutagenesis diminished but failed to abolish activity, an alternate explanation, is that these residues are peripherally involved in binding and catalysis.

NvfF discussion states that this enzyme is involved in oxidative desaturation but does not address the enzyme's role in this process. Rather suggests a mechanism for oxidation to generate an aldehyde and hemioorthoester which cyclizes subsequent to ionization to yield the orthoester. Is the proposal that this enzyme is an oxygenase and a desaturase? If so this should be made explicit and perhaps a precedent could be cited (e.g. clavaminic synthase has three functions: oxygenation, oxidative cyclization, and oxidative desaturation).

Minor comments: Add enzymes to arrows in fig 4 e,f

Summary:

This work is substantial, and the evidence is clear that several enzymes possess previously unobserved functions. The pathway is complex and its deconvolution is an impressive accomplishment. One aspect that was not discussed in this work, but a cursory literature search seems to indicate, is that novofumigatonin does not possess a known biological activity. This property renders the study a very interesting case study in mechanistic enzymology though it decreases broader interest in this work, as excellent as it is. For this reason, it recommended that publication be placed in a more specialized journal.

Response to the referees' comments

Reviewer #1 (Remarks to the Author):

The manuscript describes the elucidation of the biosynthetic pathway of the fungal natural product novofumigatonin using a combination of in vivo gene inactivations, heterologous production, and in vitro characterization of 3 enzymes. Several metabolites were identified and structurally elucidated from the genetic work, which helped lay out a clear biosynthetic scheme that is shown in Figure 2. The most interesting aspect of the manuscript is the discovery of a peroxide forming dioxygenase and an orthoester forming, iron-dependent enzyme. The experimental design is rigorous and a very impressive amount of experimental results are provided. The conclusions are strongly supported by the data. Overall this is an excellent contribution. The only considerations I have are:

1. In some regard, the manuscript is too dense as a significant portion is dedicated to results from gene inactivations of early- to middle-stage reactions, which are moreso expected than the late-stage reactions. It isn't until page 10 when the endoperoxide chemistry (formation and utilization to form the orthoester) is finally introduced. It took a considerable amount of time to work through the significance of the gene inactivation experiments in relationship to what is known for other related terpenoids. I found this somewhat distracting since the interesting part is the endoperoxide and orthoester forming chemistry. My advice would be to separate into two separate manuscripts or somehow streamline the early- and middle-stage results.

RESPONSE: We greatly appreciate the review's suggestion. We agree that our paper is somewhat dense and understand that it would take some time for readers to reach the most important point of the work. However, at the same time, we also believe that the description of the early- and mid-stages of the biosynthesis is important to understand the whole pathway of novofumigatonin and that we have provided an interesting hypothesis that the methyltransferases are required as a protecting group. In order to balance this dilemma, we have reconstructed and shortened the first three subsections in the Result section (reduced to ~80%) as highlighted in yellow, while all the key findings are retained in these subsections. We believe that this revision increased the readability of our manuscript.

2. In the first paragraph of the discussion, it is stated that their approach enabled them to "rapidly elucidate". I am not sure about the use of rapidly?

RESPONSE: We have changed “rapidly” to “efficiently” for clarity.

3. Noticed that Supplemental Figure 2 is not cited in the main text until after the introduction of several other Supplemental figures. These figures might need to be renumbered.

RESPONSE: The numbering of the Supplementary Figures was changed according to their appearances.

4. In the final, conclusion paragraph, the meaning of the statement "would even expand the catalytic versatility of the known α KG-dependent enzymes" is unclear. Potentially reword

RESPONSE: We have revised the phrase into “could even provide opportunities to engineer known α KG-dependent enzymes into novel α KG-independent biocatalysts to expand nature’s catalytic versatility”.

Reviewer #2 (Remarks to the Author):

The manuscript by Matsuda et al. described the experimental evidences for biosynthetic pathway of meroterpenoid novofumigatonin by a number of gene knockouts, heterologous gene expression in *Aspergillus oryzae* and in vitro analysis of key enzymes. Early-stage transformations to asnobolin A via farnesyl-DMOA and asnobolin H are rather standard because the similar transformations are found in biosynthesis of related meroterpenoids such as andrastin, terretonin and austinol. However, a series of oxidative transformations from asnobolin A to novofumigatonin are highly intriguing especially in view of involvement of endoperoxide 14 and its conversion to orthoester by endoperoxide isomerization. These enzymes NvfI and NvfE catalyzed the reactions that are rarely found in the literatures. This reviewer recognizes that the data presented are clear and does give interests to the readership of Nature Communication. This reviewer recommends publication following minor revisions.

1) Most of the researchers are interested in CRISPR-Cas9 based genome editing. The authors did not use this technique although they can be applied to all gene deletion experiments. This reviewer suggests to add some comments to explain the reason for this.

RESPONSE: The reason that we did not always use the CRISPR-Cas9 system is because the success rate of the gene deletion was, in most cases, good enough after the deletion of the *ligD* gene. Therefore, we only used the CRISPR-Cas9 system for the genes to which our initial attempt was not successful. To make this point clearer, we have added the following phrase in the first

paragraph of the Results section: “**which, in most cases, enabled efficient gene deletion without further usage of the CRISPR-Cas9 system and thus minimized vector construction works in the following experiments.**” (page 5, line 9-11).

2) Page 10, line 9, “hydrogenation” should be “hydrogenolysis”.

RESPONSE: Revised as suggested. Accordingly, we have also changed “Hydrogenation of fumigatonoid B (15)” to “Hydrogenolysis of fumigatonoid B (15)” in the Methods section (page 29, line 9).

3) In Figure 4, as NvfF in Figure 4g, addition of NvfI and NvfE in Figures 4e and 4f may help to differentiate these similar enzymatic reactions.

RESPONSE: Revised as suggested.

4) In Figure 5, “acetoxylation” should be “hydroxylation-acetylation”.

RESPONSE: Revised as suggested.

5) Page 11, line 5 from the bottom, “the stereochemistry of the methyl ester group ---” is not correct. It should be changed to “the stereochemistry at C5' ---”.

RESPONSE: Revised as suggested.

6) Page 12, line 5-7, the meaning of the sentence “Similarly, our attempt ----- metabolite (Fig. 2e, lane v)” is hard to understand. The phrase “the other eleven genes” should be “the other twelve genes”. Please check it.

RESPONSE: The sentence was revised to “Similarly, our attempt to isolate the genuine product from the NvfE-catalyzed reaction was not successful, as the further addition of *nvfE* and *nvfG* to the above constructed transformant only generated 18, which seems to be a shunt product, as an *nvfE*-specific metabolite” (page 11, line 10-13). The phrase “the other eleven genes” is actually correct, but to avoid confusion, this phrase was removed from the sentence.

Reviewer #3 (Remarks to the Author):

Overview:

This study by Matsuda et al describes the biosynthetic pathway of a structurally elaborate meroterpoid derived terpene novofumigatonin. The boundaries of the biosynthetic gene cluster were unambiguously determined via heterologous expression, and a proposed sequence of biochemical transformations resulting in novofumigatonin was based on a series of overlapping CRISPR-generated targeted gene disruptions, heterologous expression experiments, and in vitro turnover assays. In the process several novel transformations were uncovered including a rare endoperoxide synthase, an endoperoxide isomerase yielding an orthoester, and an orthoester isomerase interconverting orthoester constitutional isomers. The orthoester isomerase connects the biosynthesis of fumigatonin and novofumigatonin and highlights an interesting extension of the former.

The manuscript is meticulously prepared, experiments are well described, and results are statistically valid. Conclusions are drawn from experiments with adequate controls and orthogonal methods were used to validate most conclusions.

Critiques:

Regarding the mechanism of the endoperoxide forming enzyme NvfI, “a molecular oxygen is then incorporated to generate the peroxy radical 24, which undergoes C-O bond formation at C-2' to yield 25”:

There was a lack of discussion alternative mechanistic possibilities. The proposed generation of the C-13 radical is consistent with the canon, but the interception of this highly unstable species by an additional molecular oxygen to form a hydroperoxide is without precedent. Please provide precedent for the active site hosting two molecules of oxygen during a single reaction coordinate to generate a hydroperoxide.

RESPONSE: Enzymatic endoperoxidation is not an unprecedented reaction, but is known to be catalyzed by a few enzymes, such as cyclooxygenases and fomitremorgin B endoperoxidase (FtmOx1). Especially, FtmOx1 is an α -KG-dependent enzyme like NvfI, and it is proposed that FtmOx1 performs the endoperoxidation in a similar manner to that proposed for NvfI, in which a molecular oxygen is incorporated after the initial hydrogen abstraction. Thus, this reaction by FtmOx1 also requires two molecular oxygens in a single reaction, and therefore we believe that our proposed mechanism for NvfI is plausible. To make this point clear and to explain the differences between the reactions catalyzed by NvfI and FtmOx1, we have added the following sentences in the Result section: “This proposed mechanism for the endoperoxidation is similar to that proposed for the fomitremorgin B endoperoxidase (FtmOx1)^{24,25}, which is another α KG-dependent enzyme. However, the mechanisms for the radical quenching differ in these two enzymes; FtmOx1 requires a reducing agent as a hydrogen donor at the end of the reaction, while NvfI completes the reaction

by the oxygen rebound, altogether introducing three oxygen atoms to the substrate.” (pages 13, line 11-16).

Conclusions of NvfE mutagenesis seem overinterpreted. Since mutagenesis diminished but failed to abolish activity, an alternate explanation, is that these residues are peripherally involved in binding and catalysis.

RESPONSE: We agree and understand that our mutational experiment on NvfE is not sufficient to fully support the hypothesis that the four amino acid residues are all involved in the iron binding, but we would like to provide one possibility that explains the result well. We have added the following sentence in the Result section for clarity: “**However, this hypothesis should be confirmed by a future X-ray crystallographic study on NvfE complexed with iron.**” (page 16, line 12-13).

NvfF discussion states that this enzyme is involved in oxidative desaturation but does not address the enzyme’s role in this process. Rather suggests a mechanism for oxidation to generate an aldehyde and hemioorthoester which cyclizes subsequent to ionization to yield the orthoester. Is the proposal that this enzyme is an oxygenase and a desaturase? If so this should be made explicit and perhaps a precedent could be cited (e.g. clavaminic synthase has three functions: oxygenation, oxidative cyclization, and oxidative desaturation).

RESPONSE: As noticed by the reviewer, our conclusion is that NvfF catalyzes two consecutive reactions as often seen in this class of enzymes, including clavaminic synthase. We did not describe the enzyme’s role in the desaturation event since desaturation is a commonly seen reaction by α -KG-dependent enzymes, and focused on the aldehyde forming reaction, whose mechanism would not be obvious to readers. Altogether, we have modified and added sentences in the last paragraph of the Result section as follows: “**Nevertheless, it is evident that NvfF is engaged in two sequential oxidative reactions, the dehydrogenation to introduce the C-C double bond between C-1 and C-2 and the aldehyde formation at C-13. Homologues of NvfF are often found in the DMOA-derived meroterpenoid pathways, and most of them catalyze multistep reactions as NvfF^{16,22,31,32}.**

Interestingly, contrary to the commonly seen desaturation event, the aldehyde forming reaction by NvfF is relatively rare for reactions catalyzed by α KG-dependent enzymes.” (page 17, line 8-14).

Minor comments: Add enzymes to arrows in fig 4 e,f

RESPONSE: As already mentioned in the response to the Reviewer #2, the figure was revised as requested.

Summary:

This work is substantial, and the evidence is clear that several enzymes possess previously unobserved functions. The pathway is complex and its deconvolution is an impressive accomplishment. One aspect that was not discussed in this work, but a cursory literature search seems to indicate, is that novofumigatonin does not possess a known biological activity. This property renders the study a very interesting case study in mechanistic enzymology though it decreases broader interest in this work, as excellent as it is. For this reason, it is recommended that publication be placed in a more specialized journal.

RESPONSE: As pointed out, novofumigatonin unfortunately has no reported biological activity, and we agree that this could somewhat decrease the impact of our paper. Yet, we strongly believe that our paper still has a broad interest because of the following reasons.

First, α -KG-dependent enzymes are ubiquitously distributed, and they are not only involved in secondary metabolism but also play key roles in biological events in human. Despite the wide occurrence of α -KG-dependent enzymes, however, our paper reports the first example of an α -KG-independent enzyme with sequence similarity to known α -KG-dependent enzymes. Thus, this work would be of interest to a wide range of researchers studying this class of enzymes and could lead to discovery of similar enzymes that have been overlooked.

Then, we also believe that our work provides a convincing model case for biosynthetic study on natural products; although both gene deletion and heterologous expression approaches have been widely utilized, they are separately or not comprehensively performed in the previous cases. Our study indicated the importance of combining different approaches at the same stage and also demonstrated that this combined methodology can now be readily performed because of the CRISPR-Cas9 system.

Overall, we believe that our manuscript presents an important discovery and progress in broader context of chemistry and biology and that our manuscript has been improved after the revision.

REVIEWERS' COMMENTS:

Reviewer #1 (Remarks to the Author):

The authors have adequately addressed the concerns of the reviewers.

Reviewer #3 (Remarks to the Author):

The authors have adequately responded to my critiques. The most subjective of these was with regards to NCOMMS was the lack of activity of this compound and I recommended it for a more specialized publication. The authors response to this comment, in balance with the novelty of the chemistry and scope of work persuades me to agree that NCOMMS is a good venue for this work.